# Fatty Acid Profile and Lipid Quality Indexes of the Meat and Backfat from Porkers Supplemented with EM Bokashi Probiotic

**DOI:** 10.3390/ani13203298

**Published:** 2023-10-23

**Authors:** Zuzanna Goluch, Artur Rybarczyk, Ewa Poławska, Gabriela Haraf

**Affiliations:** 1Department of Food Technology and Nutrition, Wrocław University of Economics & Business, ul. Komandorska 118-120, 53-345 Wrocław, Poland; zuzanna.goluch@ue.wroc.pl (Z.G.); gabriela.haraf@ue.wroc.pl (G.H.); 2Department of Animal Nutrition and Feed Science, Wrocław University of Environmental and Life Science, Chełmońskiego 38C, 51-630 Wrocław, Poland; 3Institute of Genetics and Animal Biotechnology of the Polish Academy of Sciences, Postępu 36A, 05-552 Magdalenka, Poland; e.polawska@igbzpan.pl

**Keywords:** probiotic, meat, pork, backfat, fatty acid, lipid quality indexes

## Abstract

**Simple Summary:**

Probiotics can be used in pigs fattening as a partial or full alternative to antibiotic growth promoters. Their composition and the dosage used in the feed must not impair the nutritional value of the meat raw materials obtained from them. The use of the probiotic preparation Bokashi (containing specific strains of *Lactobacillus* and *Saccharomyces cerevisiae* yeast) as a feed additive in fattening pigs resulted in significant changes in meat and backfat fatty acid profiles and the lipid indexes calculated from them. The changes in the fatty acid profile of backfat appeared to be more beneficial to consumer health due to the increased proportion of polyunsaturated fatty acids (PUFA). From a technological point of view, there is a more favorable Saturation Index (SI) value in backfat.

**Abstract:**

The study aimed to assess the effect of supplementation of pig diet with the Bokashi probiotic on the fatty acid profile of *longissimus lumborum* (*LL*) muscles and backfat. The research involved 120 hybrid pigs deriving from Naïma sows and P-76 boars. The experimental group’s pigs received probiotics in their feed (containing *Saccharomyces cerevisiae*, *Lactobacillus casei*, and *Lactobacillus plantarum*). To analyze the fatty acid profile in intramuscular fat (IMF) of *LL* and backfat, 24 pig carcasses from the control group and 26 from the probiotic-supplemented group were randomly selected. Probiotic supplementation increased the Atherogenic Index, reduced the proportion of C20:4, and increased C12:0 and C18:2 n-6 in IMF *LL*, without affecting ΣSFA, ΣMUFA, and ΣPUFA. In backfat, probiotic supplementation decreased C18:1 and C18:2 n-6 proportion and increased C18:3 n-3, C20:3 n-6*,* and C20:4 n-6. These changes resulted in significantly higher ΣMUFA, ΣPUFA, PUFA Σn-3/Σn-6, and lower Saturation Index (SI). From a consumer health and technological point of view, probiotic supplementation improved the lipid profile of backfat to a greater extent than *LL* muscle. Bokashi, at a dose of 3 g/kg of feed in the last stage of pig production, had no significant effect on the fatty acid profile of the meat.

## 1. Introduction

Due to well-established habits, culinary traditions, culinary tourism, nutritional value, and consumer affordability, pork and its products are the components of people’s diets in many countries [1,2,3]. Moreover, pork is the most frequently and most consumed meat globally [4]. The UN’s Food and Agriculture Organization predicts that pig farming is the animal industry that will experience the strongest growth, with an expected increase of 8.6% by 2030 and 12.7% by 2050 [5].

One of the consumer concerns about the health safety of pork is the use of antibiotic growth promoters (AGPs) by farmers, although a ban on their use in pig fattening was introduced in the European Union in 2006, in the U.S. in 2017, and in China in 2020 [6,7,8]. Reports continue to appear in the press about their detection in pork carcasses in countries [9,10] that lack regulations requiring testing for antibiotic residues in meat destined for the local market. Unfortunately, AGPs can cause cross-resistance induction of pathogenic bacterial strains in humans and the possibility of allergic reactions [11,12,13]. Also, a problem in animal fattening is the unjustified and excessive use of therapeutic antibiotics [14], which can cause antimicrobial resistance (AMR). On the other hand, AMR is a serious threat to animals and humans [15,16]. The WHO has announced that AMR is one of the top 10 global public health threats facing humanity. It is estimated that by 2050, about 10 million people will die each year from AMR if action is not taken to address the problem through a One Health approach [17]. The agri-food sector must take action to counter this phenomenon, among other things, by practicing sustainable agriculture. The World Organization for Animal Health (OIE) developed standards on “Monitoring of the quantities and usage patterns of antimicrobial agents used in food-producing animals”. In the Global Action Plan on AMR framework, the OIE has also built a global database on antimicrobial agents for animal use, supported by the Tripartite collaboration (WHO, FAO, and OIE) [18].

Using probiotics instead of antibiotic growth promoters increases the health quality of meat. According to a study (conducted among 11,294 consumers of pork meat from 14 countries in Europe, America, and Asia), the health quality of pork is a very important determinant of purchase [1]. This factor was cited first, ahead of sensory quality, price, and animal welfare. The first preferences regarding pork production were also quality and health, followed by environmental and animal friendliness, regional identity, and production efficiency.

In 2020, the European Union developed an action plan for Europe’s “Green Deal”, one of the components of which is the strategy of sustainable agriculture “from farm to fork” for a fair, environmentally friendly, and healthy food system. The strategy makes clear that there is an urgent need to reduce reliance on antimicrobials and improve animal welfare, among other things [19]. Currently, the therapeutic and prophylactic use of antibiotics in animal production remains legal [20]. Within the framework of Sustainable Food Systems [21], livestock farmers are looking for alternative natural feed additives that can positively affect the health and welfare of animals and the quality and functional properties of the meat they obtain from them [22,23]. Such feed additives, alternatives to AGP, used in pig fattening include direct-fed microbials (DFMs), more widely known as probiotics.

DFMs are categorized into three main groups: *Bacillus*, lactic acid-producing bacteria, and yeast [24]. DFMs improve the immune response, maintain the eubiosis state of the animal’s intestines, and thus modulate their health and improve their performance at all stages of production [23,25,26,27,28,29]. Extensive reviews on the benefits of probiotics in swine production have been presented by Liu et al. [24], Dewulf et al. [14], and Pereira et al. [30]. Despite the relationship between probiotics and health described in the literature, progress in research on their impact on animal production is still insufficient.

Scientific studies have paid little attention to the effect of probiotics in pig fattening on meat’s energy and nutritional value. Present-day consumers are very conscious of health and nutrition issues. Thus, they demand healthier meat products. In pro-health nutrition education, consumers are recommended to remove visible external fat from the meat to improve the blood’s lipid profile and reduce body weight. Backfat, however, is traditionally eaten in many societies directly or used as a frying medium (lard) for several food products mainly used in domestic cooking [31,32]. Therefore, foods’ fat content and fatty acid profile are essential in dietary prophylaxis and diet therapy for non-communicable diseases [33,34].

The study aimed to assess the effect of supplementation of pig diet with the probiotic EM Bokashi on the fatty acid profile of intramuscular fat (IMF) of their muscles and backfat, as well as on lipid quality indicators informing about the possible impact on consumer health.

## 2. Materials and Methods

### 2.1. Animals, Feed, Slaughter, and Meat Sampling

In our study, we analyzed the changes in the fatty acid profile of pork and pork fat due to using the probiotic EM Bokashi. Its dosage was developed and improved by the breeder to provide the best results in terms of productivity and production economics. The research was conducted with 120 commercial hybrid pigs (P-76 boars and Naïma sows) raised on a Pomeranian Voivodship (Poland) production farm. The experiment was carried out in one building equipped with a gravity ventilation system. The fattening pigs were kept on deep straw litter. Moreover, during the fattening period, from the 1st to the 12th week, the ambient temperature gradually decreased from 19 °C to 16 °C. The fattened pigs received feed and water ad libitum. The age at the end of fattening is 162 days. During the growth period, the pigs (aged 28 to 164 days) were raised under the same environmental conditions and were fed ad libitum the same balanced, dry, loose, complete feed. Detailed information about diet components, chemical composition of feed, and fattening conditions was previously published [28].

The pigs in the control group (n = 60) were fed the feed without the addition of probiotic EM Bokashi. Pigs from the experimental group (n = 60) were fed the feed supplemented with the EM Bokashi probiotic (wheat bran, sugar cane molasses (0.0785 mL/100 g), a complex of probiotics, e.g., *Saccharomyces cerevisiae* IFO 0203 3.3 10^5^ cfu/g; *Lactobacillus ca*s*ei* ATCC^®^7469™ 1.95 10^7^ cfu/g; *Lactobacillus pla*n*tarum* ATCC^®^8014™ 1.95 10^7^ cfu/g). The probiotic was produced by Greenland Technologia EM^®^ (Janowiec n/Wisła, Poland), an authorized representative of EM Research Organization (EMRO) Japan Technology in Poland. The probiotic addition depended on the age of the piglets, i.e., from the 28th day of life to reach a body mass of 12 kg was 10 g/kg; up to a body mass of 20 kg was 7 g/kg; up to a body mass of 30 kg was 5 g/kg; during the fattening period from 30 kg to the end of rearing was 3 g/kg. This dosage of probiotic allowed pig producer to improve the biological efficiency of production while maintaining the profitability of production. No antibiotics were administered to pigs in the experimental and control groups for therapeutic purposes. After reaching a body weight of about 112 kg, the pigs were brought to the meat plant and rested for about 16 h. They were slaughtered the next morning, after a cumulative pre-slaughter fasting period of 33 h and 30 min. Animals were slaughtered using Butina’s CO_2_ gas stunning system in a commercial slaughterhouse (Marel, Garðabær, Iceland).

A two-stage blast and conventional chilling system were used for carcass chilling. After evisceration, carcasses were cooled in a blast-cooling tunnel at −24 °C for 70 min. Next, the carcasses were placed in an equilibration cooler at 1 °C for 22 h. Experimental material (*longissimus lumborum muscle* (*LL*) and backfat) was obtained from 26 carcasses of a similar weight (HCW: 85 ± 5 kg) and sex (1:1) randomly chosen from the probiotic-supplemented group and 24 carcasses selected from the control group. The *longissimus lumborum* (*LL*) muscle samples containing backfat layers were collected between the right half-carcasses’ 1st and 4th lumbar vertebral regions. The samples were packed in vacuum polyethylene bags and transported to the laboratory in thermos within one hour. Immediately upon delivery to the laboratory, the *LL* muscle was separated from the fat and bones. Next, all *LL* muscle samples and backfat were cut into 4 cm thick slices (starting from the cranial end), placed in polyethylene bags, and frozen at −80 °C for one month until fat content and fatty acids determinations.

### 2.2. Crude Fat Content of Meat

The crude fat content in *LL* muscle was analyzed by the methods of the Association of Official Agricultural Chemists (AOAC) [35], with use of a Soxtec HT6 by Foss Tecator (Hillerød, Denmark) (960.39 (a), p. 39.1.05) by petroleum ether extraction.

### 2.3. Fatty Acid Analysis

The fatty acid profiles of *LL* muscle intramuscular fat and backfat were determined by gas chromatography with the AGILENT Tech. 7890A Chromatograph, equipped with a flame-ionization detector (FID). Earlier ground meat samples were homogenized with a T 25 Ultra Turrax (IKA-Werke GmbH & Co. KG, Staufen, Germany). Fat extraction was performed using the Folch et al. [36] method, and the fatty acids methyl esters were formed according to the 996.06 method of AOAC [37], using SOCl_3_ in methanol instead of BF3. According to the method, samples were homogenized using chloroform: methanol (2:1; *v*/*v*) solution. The resulting methyl esters of fatty acids (FAMEs) were analyzed on the 60 m Hewlett-Packard-88 capillary column (Agilent J&W GC Columns, Santa Clara, CA, USA)—100 m × 0.25 mm × 0.20 mm film thickness. The detector and injector temperature were 260 °C. The injection volume was 1.0 mL and a split ratio 1:40. Helium was a carrier gas and the flow rate was 50 mL/min. The initial column temperature (140 °C) was held for 5 min and then programmed to increase at a rate of 4 °C/min to 190 °C and next to 215 °C at a rate of 0.6 °C/min. The analysis lasted 61 min. Identification of FAMEs was carried out by comparing their retention times with external standard (Supelco 37 FAME Mix C4–C24 Component, Sigma Aldrich, St. Louis, MO, USA). The internal standard used to control retention times was C 11:0 acid (cat no. 89764, Fluka, Sigma Aldrich, St. Louis, MO, USA). Samples were analyzed in four repetitions. The fatty acids were presented as a percentage (*w*/*w*) of total fatty acids.

### 2.4. Reagents

Methanol (cat no. 1.06018), chloroform (cat no. 34854), acetic acid from Merck, and SOCl_3_ from Sigma Aldrich were used. Double-distilled water was purchased from a Milli-Q Water System (Millipore, Billerica, MA, USA). Fatty acids standards were obtained from Sigma Aldrich, Neochema (Bodenheim, Germany), and Cayman (Ann Arbor, MI, USA).

### 2.5. Lipid Quality Indexes

Nowadays, more and more indexes are being calculated from foods’ fatty acid profiles to determine their lipids’ pro- or anti-health nature (e.g., ΣMUFA/ΣSFA, ΣPUFA/ΣSFA, ΣUFA/ΣSFA). In addition, indicators that determine the process of fatty acid metabolism involving desaturases and elongases are calculated. (e.g., Δ^9^-desaturase index, Total Desaturation Index, Elongation Index). The values of some lipid indexes (e.g., Index of Atherogenicity, Index of Thrombogenocity, hypocholesterolemic fatty acids/hypercholesterolemic fatty acids ratio, Health Promoting Index) correlate with the risk of developing non-communicable diseases, which is essential for the consumer from a nutritional point of view. To assess the nutritional value of food lipids, lipid indexes such as the Nutri Value Index or Nutritional Ratio can be used. On the other hand, indexes such as the Unsaturation Index and Peroxidation Index provide information on the lipid profile, susceptibility to oxidation, and, thus, stability during storage, which is essential from a technological point of view [38].

For example, in the diet for the prevention of ischemic heart disease (IHD), it is recommended that the ΣPUFA/ΣSFA ratio be higher than 0.45 [39] or at least 0.4 [40]. The higher a product’s ΣPUFA/ΣSFA ratio, the greater its potential health benefits. Foods with a value of this index lower than recommended can be considered undesirable for the human diet because of their potential to increase cholesterol in the blood. Index of Atherogenicity represents the relationship between hypercholesterolemic (favoring the adhesion of lipids to cells of the immunological and circulatory system) and protective fatty acids (inhibiting the aggregation of plaque and diminishing the levels of esterified fatty acid, cholesterol, and phospholipids, thereby preventing the appearance of micro- and macro-coronary diseases) [40].

Table 1 shows grouped fatty acids and lipid quality indexes concerning human health, also determining the nutritional value of the tested food and its lipids.

### 2.6. Statistical Analysis

Statistical analysis included one-way analysis of variance (ANOVA). The significance of differences between groups was assessed by Tukey’s test at *p* ≤ 0.05 and *p* ≤ 0.01 levels of significance, using Statistica^®^13.1 software [51]. Findings were presented as arithmetic means with standard deviations (SD).

## 3. Results and Discussion

In recent years, attention has been drawn to the vital role of gut microbiota composition on the health of livestock, their welfare, and the quality of meat obtained from them. Prevention of intestinal dysbiosis through the therapeutic use of probiotics and prebiotics in rearing and fattening is one of the forms of improving immunity and reducing animal morbidity and mortality [52]. Equally important is to determine the effect of dietary supplementation of slaughter animals with probiotics on the quality and functional properties of their meat, sensory characteristics, microbiological safety, and shelf life [25,41,53]. It is generally known that the fat content in pork and its fatty acids profile are influenced by, among others, the following factors: sex, age, breed, body ratio, protein ratio, energy in feed, feeding method, energy consumption, and type of fat, the composition of the intestinal microbiotas, well as the efficiency of the metabolism of fatty acids in the body, with intraindividual variations due to genetic disposition [54,55]. The health quality of pork lipids can be improved through proper feeding of pigs, as there is a correlation between the fatty acid profile of feed and the fat tissue profile of the animal [56]. However, consumers’ attitudes toward fat content in pork can be contradictory [57]. On the one hand, they perceive high-fat content in meat negatively. Simultaneously, its presence is welcomed because it is associated with better taste, tenderness, aroma, and juiciness [57]. The effect of probiotic organisms on the production of fatty acids has already been described in the literature. In the gastrointestinal tract (GIT), the fermentation of indigestible carbohydrates that occurs there (with the participation of probiotic and commensal bacteria) produces CO_2_, H_2_, and CH_4_, as well as acetate, propionate, and butyrate short-chain fatty acids (SCFAs) [58]. Through the blood route, acetate is transported to muscle and other tissues and metabolized. The liver captures propionate from the circulation, which participates in gluconeogenesis. The synthesized glucose in the glycosylation process is converted to pyruvate, which is decarboxylated, and acetyl Co-A is formed. It is used in the Krebs cycle to synthesize ATA but can also participate in de novo fatty acid synthesis (liponeogenesis), cholesterologenesis, and/or ketogenesis [59]. The resulting fatty acids can undergo esterification to triacylglycerols to form a pool of lipids in muscle, liver, or fat tissue [60,61]. Thus, through this mechanism, probiotic bacteria used in animal fattening can affect the lipid content of muscle and adipose tissue. SCFAs are metabolites produced by gut bacteria that can regulate host metabolism. Most SCFAs produced in the intestine are absorbed by the host, contributing to its energy. Studies have found the beneficial effects of SCFAs on immune regulation by suppressing the production of pro-inflammatory cytokines, such as IL-6 while increasing IL-10 secretion [62].

### 3.1. Fat Content and Fatty Acids Profile in Longissimus Lumborum Muscle of Pigs

Our study showed no significant (*p* > 0.05) differences in crude fat in *LL* between individuals in the control group and the probiotic-supplemented group EM Bokashi (Table 2). Similarly, Rekiel et al. [63] showed no significant effect of the Bactocell probiotic (contains bacteria of the strain *Pediococcus acidilactici* MA 18/5M) in fattening pigs (crossbreeds of Polish Large White × Polish Landrace coming from Duroc and Belgian Landrace sires) on the fat content of the *longissimus dorsi* muscle (*LD*), compared to the control group. Likewise, Parra et al. [64] did not establish a significant impact of feed supplementation with Bioplus 2B (*Bacillus licheniformis* and *B. subtilis* mixture) in Iberian pigs on the fat content in *serratus ve*n*tralis (SV)* muscles. Also, Goluch et al. [41] showed no effect of dietary supplementation of fattening pigs (Landrace–Yorkshire × Duroc) with Bio-Plus YC probiotic (containing *Bacillus licheniformis* DSM 5749 and *Bacillus subtilis* DSM 5750) on crude fat content in *LL*.

EM Bokashi contained *Saccharomyces cerevisiae* and LAB. In other studies conducted in pigs ((Landrace × Yorkshire) × Duroc), which were given this yeast at 0.2% or 0.3% of feed, there were also no significant differences in fat content compared to the control group [65].

It has long been known that an increase in the consumption of SFA by humans correlates (due to a rise in serum total cholesterol levels) with the risk of cardiovascular disease (CVD) [66] and cancer [67,68]. Supplementation of the pig diet with the probiotic EM Bokashi significantly (*p* ≤ 0.05) increased the proportion of lauric acid (C12:0) (Table 1). However, an increase in the C12:0 content of IMF *LL* muscle did not significantly change the total percentage of SFA (Table 2). In addition, arachidic (C 20:0) and behenic (C22:0) acids were not determined in the probiotic-supplemented group. It is a positive development, as long-chain saturated fatty acids LCSFA (C12–18, especially palmitic acid) have been shown in cohort studies to increase CVD and cancer risk [69]. In contrast, short-chain fatty acids SCFA and medium-chain fatty acids MCFA (C4–C10) may be more beneficial or neutral for the consumer’s health. Lauric acid has been found to exhibit antiproliferative and proapoptotic effects in human breast and colon cancer cells [70,71], reduce the risk of developing CVD [72], or be neutral in this regard [73]. Similarly, Tufarelli et al. [74] did not observe a significant difference in SFA proportion in IMF of *longissimus dorsi* (*LD*) of pigs ((Landrace × Yorkshire) × Duroc) that were supplemented with SLAB51 probiotic containing *Streptococcus thermophilus*, *Bifidobacterium animalis* ssp. *lactis*, *Lactobacillus acidophilus*, *Lactobacillus helveticus*, *Lactobacillus paracasei*, *Lactobacillus pla*n*tarum*, and *Lactobacillus brevis*). Also, Parra et al. [64] did not establish a significant impact of feed supplementation of Iberian pigs with Bioplus 2B probiotic on changes in percentage SFA in IMF of *SV*. Rekiel et al. [63] showed no significant effect of the use of the probiotic Bactocell in fattening pigs on the proportion of SFA in the IMF *LD* acid profile compared to the control group. On the other hand, Ross et al. [75] stated that the proportion of SFA in *LD* muscle was significantly lower (*p ≤* 0.05) as a result of supplementation with *L. amylovorus* and *Enterococcus faecium* mixed culture (10^8^ CFU/mL) probiotics in fattening pigs. Goluch et al. [41] found that the IMF *LL* of the pigs supplemented with BioPlus YC was characterized by a significantly lower (*p* ≤ 0.05) proportion of the C10:0 in comparison to the control group.

Considering MUFA, the effect of using probiotic supplementation EM Bokashi was marked by a significant (*p* ≤ 0.01) reduction in the heptadecanoic acid (C17:1 n-7) proportion in the IMF *LL* fatty acid profile compared to the control group. Cis-Vaccenic acid is a C18:1 n-7 isomer of oleic acid (C18:1 n-9). It is synthesized from palmitic acid (C16:0) via the production of C16:1 n-7 by an Δ^9^ desaturase and elongation by an elongase giving C18:1 n-7 [47]. The Δ^9^ desaturase, known as SCD, converts SFA into MUFA. The primary role of ∆^9^ desaturase (1 stearoyl-CoA, SCD1) in animal bodies is to limit the availability of palmitic acid (C16:0) by its conversion into oleopalmitic acid (C16:1 n-7), and thus to provide fluidity and permeability to the biological membrane. Another method to limit the presence of palmitic acid is to accelerate its elongation into stearic acid (C18:0) and its desaturation into oleic acid (C18:1 n-9). The C18:1 n-9, as the basic product of ∆^9^desaturase, is also the main fatty acid in triacylglycerols of mammals, which is used to synthesize phospholipids and cholesterol esters [76]. This continuity of conversion is noticeable in the supplemented animal group, as there is more C18:1 n-9 acid than in the control group, although this difference is not statistically significant (*p* > 0.05). Thus, converting C18:1 n-7 acid to C18:1 n-9 is beneficial to consumer health, as it is a critical factor in preventing CVD by contributing to lowering serum total cholesterol levels while increasing its antiatherogenic HDL fraction.

Table 2 shows a slightly higher content of PUFA n-3 (*p* > 0.05) acids in the supplemented group, while in the control group, ΣPUFA n-6. The activity of Δ^6^desaturase involves, among other things, the conversion of C18:2 n-6 acid to C18:3 n-6 and is also involved in the transformation pathway of C18:3 n-3 to C 22:6 n-3 acid. In the IMF *LL* fatty acid profile, there was a significantly (*p* ≤ 0.05) higher proportion of C18:2 n-6 and lack of C18:3 n-6 acid in the probiotic-supplemented group EM Bokashi (Table 2). It suggests a lower activity of Δ^6^desaturase in the PUFA n-6 acid metabolism pathway and a higher activity in the PUFA n-3 acid metabolism pathway in the supplemented animals compared to the control group. It is also confirmed by the presence of docosahexaenoic acid (C22:6 n-3) in the IMF *LL* fatty acid profile of the supplemented group and its absence in the IMF *LL* of the control group. In the control group, on the other hand, a higher proportion of C20:4 n-6 acid was found, confirming the action of Δ^6^ desaturase in the pathway of PUFA n-6 acid metabolism in this group of animals.

However, the observed changes in the share of single fatty acids did not affect the total amount of PUFA, which did not differ significantly between groups of animals. However, it is worth emphasizing the fact of a significant reduction in the *LL* lipids of the group supplemented with EM Bokashi, a precursor of pro-inflammatory eicosanoids, i.e., arachidonic acid, and the presence of DHA acid, a precursor of anti-inflammatory eicosanoids [77]. It indicates the low activity of Δ^6^-desaturase and Δ^5^-desaturase responsible for the conversion of C18:2 n-6 by C18:3 n-6 to C20:4 n-6. The anti-inflammatory properties of PUFAs n-3 may help prevent atherosclerosis, plaque rupture, and cardiovascular mortality. Chronic inflammation is a characteristic of several disorders, including diabetes and cardiovascular disease (CVD).

Studies by other authors have found different effects of dietary supplementation with probiotics on the ΣPUFA content of pig IMF. Parra et al. [64] showed no effect of feed supplementation with the probiotic BioPlus 2B on changes in the share of ΣPUFA in the IMF of SV Iberian pigs. However, Ross et al. [75] confirmed (*p* ≤ 0.05) an increase in C18:2 n-6 and C18:3 n-3 acids content in IMF *LD* of pigs supplemented with *L. amylovorus* and *Enterococcus faecium* mixed. Tufarelli et al. [74] found an increased (*p* ≤ 0.05) content of ΣPUFA in IMF *LD* of pigs supplemented with SLAB51 probiotic in comparison with the control group. Chang et al. [25] indicated a higher (*p* ≤ 0.05) content of ΣPUFA in IMF *LD* of pigs supplemented with a probiotic that contained *Lactobacillus pla*n*tarum*. A decrease in the level of eicosatrienoic acid (C20:3 n-3), with a simultaneous decrease in ΣPUFA n-3, was found by Goluch et al. [41] in LL pigs fed with the BioPlus YC probiotic.

### 3.2. Lipid Quality Indexes of Longissimus Lumborum Muscle of Pigs

In our study, we found no significant differences in the values of ΣPUFA/ΣSFA between the groups of porkers. Still, they were close to recommended values, which benefits the consumer. Parra et al. [64] and Goluch et al. [41] also showed no significant effect of probiotic feed supplementation on changes in ΣPUFA/ΣSFA values in pig intramuscular fat. On the other hand, Ross et al. [75] observed significantly increased values of this ratio in the IMF *LD* of fattening pigs fed with feed supplemented with the *L. amylovorus* and *Enterococcus faecium* probiotic mixed culture.

Additionally, attention is paid to the n-3 PUFA/n-6 PUFA ratio to predict the impact of food on health. In pork, the ratio is skewed toward n-6, which is not in line with current dietary recommendations. Dietary n-3*/*n-6 PUFA ratios ranging from 0.25 to 1 decreased the risk of breast, prostate, colon, and renal cancers [78]. In our study, the value of this index was not significantly different between the lipids of the two groups of animals and was lower than recommended.

The confirmation of the lack of significant differences in the sums of SFA, MUFA, and PUFA acids between the groups of tested animals is the similar value of fatty acid conversion indexes, i.e., desaturation DI(16), DI(18), TDI, and elongation (EI). Generally, the PUFAs n-3 play a significant role in regulating the thrombogenic index, while n-6 is dominant in the atherogenic ones. A healthy animal product can be characterized by low atherogenic and thrombogenic indexes [79]. Regarding human health, Ulbricht and Southgate [40] recommended that the atherogenicity index (AI) should be lower than 0.5. Of the indexes calculated in the present study (Table 3), significant differences (*p* ≤ 0.05) were observed only in the value of the AI index. Its higher value was observed in the group of porkers supplemented with the probiotic EM Bokashi compared to the control group. It was due to the lower (Table 2) summed proportion of MUFAs and PUFAs and higher C12:0 and C16:0 in the IMF *LL* of the control group. Thus, based on the obtained values of the index, it can be concluded that despite the effect of the probiotic on increasing the proportion of single unsaturated fatty acids in meat IMF, their impact on the consumer body may be suppressed by a higher proportion of SFA. Unfortunately, in the IMF *LL* of both groups of animals, the value of this index also exceeded the recommended 0.5. Similar results were obtained by Goluch et al. [41]. On the other hand, Ross et al. [75] showed (*p* ≤ 0.05) a lower AI value in IMF *LD* of fattening pigs under the influence of the *L. amylovorus* and *Enterococcus faecium* mixed culture used in nutrition, compared to the control group, which they explained by a decrease in ΣSFA and a significant increase in CLA (conjugated linoleic acid) content.

The Thrombogenicity Index was defined as the relationship between the pro-thrombogenic (saturated) and the anti-thrombogenic fatty acids [40]. The TI indicates the tendency to form clots in the blood vessels. Regarding human health, the value of TI is recommended to be lower than 1.0 [80]. Our studies found no significant differences in the value of this index between the lipids of both animal groups, and they were lower than recommended. However, the TI value was more favorable for the consumer’s health in the muscle lipids of the porkers from the control group. The values of the other calculated indexes of IMF *LL* lipid nutritional value (NAVI, HPI, and NR), stability, and susceptibility to oxidation (SI, UI, and PI) did not differ significantly between groups of animals and were similar to each other, indicating that the use of probiotic does not change anything in this regard. Similar results were obtained by Goluch et al. [41].

### 3.3. Fatty Acids Profile in Backfat of Pigs

Animal fats are tissue fats that can be obtained from various animals. Lard is fat extracted from the adipose tissue of a pig [81]. It is used for deep-frying and to produce margarine, shortenings, and sausages added to food products to reduce production costs [82]. The fatty acid profile of lard depends on the breed, sex, and slaughter weight of the pigs, among other factors [83]. In our study, there were no significant differences between the groups of test animals in terms of SFA (Table 4). However, the effect of probiotic supplementation EM Bokashi was marked by a significant (*p* ≥ 0.01) reduction in the proportion of C18:1 cis n-9, which translated into differences in ΣMUFA between groups of animals. In our own research previously published, we found also that the use of the Bokashi probiotic significantly (*p* ≤ 0.01) reduced the backfat thickness in fattening pigs, compared to those from the control group (16.54 vs. 19.64 mm) [28].

The supplemented group of fattening pigs also showed a significant (*p* ≤ 0.01) increase in the C18:3 n-3 acid proportion compared to the control group. It resulted in a significant rise in ΣPUFA n-3. However, within PUFA n-6, an increase in C18:2 n-6 *cis* and C20:3 n-6 and C20:4 n-6 was observed (*p* ≤ 0.01), translating into a significant rise in ΣPUFA n-6. It confirms the sequence of conversion of linoleic acid under Δ^6^-desaturase to γ-linolenic acid, then with the participation of Elovl-5 elongases to dihomo-γ-linolenic acid and further under Δ^5^-desaturase to arachidonic acid [47,84]. As shown [85], increasing the content of γ-linolenic acids and linoleic acid can increase the transparency (which is related to the extent of solidification of the fat) and softness of backfat, but these characteristics may not be acceptable to the consumers.

The proportion of PUFA acids in the fatty acid profile of backfat was significantly (*p* ≤ 0.01) higher in the probiotic-supplemented pig group than in the control group. From a technological point of view, this phenomenon is not entirely beneficial. Soft fat is characterized by an unsightly appearance, difficulty cutting, and a more rapid tendency toward oxidative rancidity than hard fat. Unsaturated fatty acids (UFA) have lower melting points than the associated SFA, so fat tissue with a higher proportion of UFA will be less solid and more translucent at a given temperature [85]. From a nutritional standpoint, increasing the proportion of pro-inflammatory arachidonic acid in backfat is unsuitable for health.

Considering the value of lipid indexes in the backfat (Table 5), there was a significant (*p* ≤ 0.01) effect of dietary supplementation with the probiotic EM Bokashi on the increase in ΣPUFA/ΣSFA values. This is positive for the consumer, as this group’s mean of 0.44 is close to the recommended value for preventing ischemic heart disease (0.45). From a nutritional point of view, the significantly (*p* ≤ 0.05) higher value of the PUFA Σn-3/Σn-6 index in the backfat of probiotic-supplemented porkers is also beneficial for the consumer. Also, a significantly (*p* ≤ 0.01) lower value of SI (Saturation Index) is advantageous from a technological point of view. The values of the other calculated indexes of dietary value of lipids (NAVI, HPI, and NR), stability, and susceptibility to oxidation (UI, PI) did not differ significantly between groups of animals.

Since no similar work was found on the effect of probiotic use on the fatty acid profile and lipid indexes of pork backfat, a broader discussion of the results obtained is limited.

## 4. Conclusions

In our opinion, EM Bokashi probiotic at a dose of 3 g/kg of feed in the last stage of pig production does not have a significant effect on the fatty acid profile of the meat. A more pronounced effect of the probiotic EM Bokashi, used in pig fattening, on the fatty acid profile and values of lipid indexes was marked in backfat than in the IMF of *longissimus lumborum* muscle. The changes in the fatty acid profile and lipid indexes in backfat are more favorable to consumer health than meat. Since consumers more often consume meat than pork fat, it is reasonable to conduct further research on other doses of this probiotic. However, further research is needed to evaluate the impact of using the Bokashi probiotic on organoleptic characteristics, consumer acceptability, susceptibility to oxidation during storage, and suitability of backfat for use in culinary processes.

## Figures and Tables

**Table 1 animals-13-03298-t001:** Calculation of individual fatty acids groups and lipid indexes.

Fatty Acids Group/Indexes	Name	Calculation Formula	References
ΣSFA	Saturated fatty acids	Sum from C4:0 to C24:0	[41]
ΣMUFA	Monounsaturated fatty acids	Sum from C14:1 to C24:1	[41]
ΣPUFA n-3	Polyunsaturated fatty acids n-3	C18:3 n-3 + C18:4 n-3 + C20:3 n-3 + C20:5 n-3 + C22:5 n-3 + C22:6 n-3	[41]
ΣPUFA n-6	Polyunsaturated fatty acids n-6	C18:2 n-6 + C18:3 n-6 + C20:3 n-6 + C20:4 n-6 + C22:2 n-6 + C22:4 n-6	[41]
ΣPUFA	Polyunsaturated fatty acids	Σ n-3 PUFA + Σ n-6 PUFA	[41]
ΣUFA	Unsaturated fatty acids	ΣMUFA + ΣPUFA	[41]
ΣMUFA/ΣSFA	Monounsaturated/Saturated fatty acids		
ΣPUFA/ΣSFA	Polyunsaturated/Saturated fatty acids	Σ n-3 PUFA + Σ n-6 PUFA/ΣSFA	[40]
ΣUFA/ΣSFA	Unsaturated/Saturated fatty acids	ΣMUFA + ΣPUFA//ΣSFA	[41]
ΣPUFA n-3/ΣPUFA n-6	Polyunsaturated fatty acids n-3/Polyunsaturated fatty acids n-6	C18:3 n-3 + C18:4 n-3 + C20:3 n-3 + C20:5 n-3 + C22:5 n-3 + C22:6 n-3/C18:2 n-6 + C18:3 n-6 + C20:3 n-6 + C20:4 n-6 + C22:2 n-6 + C22:4 n-6	[42]
AI	Atherogenicity index	[C12:0 + (4 × C14:0) + C16:0 + C18:0]/[Σ MUFA + Σ PUFA n-6 + Σ PUFA n-3	[40]
TI	Thrombogenicity index	(C14:0 + C16:0 + C18:0)/[(0.5 × Σ MUFA) + (0.5 × Σ PUFA n-6) + (3 × Σ PUFA n-3) + (Σ PUFA n-3/Σ PUFA n-6)]	[40]
SI	Saturation Index	(C14:0 + C16:0 + C18:0)/(Σ MUFA cis + Σ PUFA).	[41]
UI	Unsaturation Index	1 × (% monoenoics) + 2 × (% dienoics) + 3 × (% trienoics) + 4 × (% tetraenoics) + 5 × (% pentaenoics) + 6 × (% hexaenoics)	[43]
h/H	hypocholesterolemic fatty acids/Hypercholesterolemic fatty acids ratio	[C18:1 cis n-9 + C18:2 n-6 + C18:3 n-6 + C18:3 n-3 + C20:3 n-6 + C20:4 n-6 + C20:5 n-3 + C22:4 n-6 + C22:5 n-3 + C22:6 n-3)]/(C14:0 + C16:0)	[44]
PI	Peroxidisability Index	(% monoenoic acid × 0.025) + (% dienoic acid × 1) + (% trienoic acid × 2) + (% tetraenoic acid × 4) + (% pentaenoic acid × 6) + (% hexaenoic acid × 8)	[45]
DI (16)	Δ^9^-desaturase Index activity for 16:0	100 [C16:1 n-9/(C16:1 n-9 + C16:0)]	[46]
DI (18)	Δ^9^-desaturase Index activity for 18:0	100 [C18:1 n-9/(C18:1 n-9 + C18:0)]	[46]
TDI	Total Desaturation Index	(C16:1 n-7 + C18:1 n-7 + C18:1 n-9)/(C14:0 + C16:0 + C18:0)	[47]
EI	Elongation Index	100 [(C18:0 + C18:1 n-9)/(C16:0 + C16:1 + C18:0 + C18:1 n-9)]	[47]
NVI	Nutritive Value Index	(C18:0 + C18:1 n-9)/C16:0	[48]
HPI	Health Promoting Index	ΣSFA/[C12:0 + (4 × C14:0) + C16:0]	[49]
NR	Nutritional ratio	(C12:0 + C14:0 + C16:0)/(C18:1 n-9 + C18:2 n-6)	[50]

**Table 2 animals-13-03298-t002:** Fat content and fatty acid profile (% of all fatty acids) of *longissimus lumborum* muscle.

Fat and Fatty Acids	Control	Bokashi	*p*-Value
(%)	(n = 24)	(n = 26)	
Crude fat	1.42 ± 0.09	1.46 ± 0.08	0.722
C12:0	0.53 ^b^ ± 0.98	1.19 ^a^ ± 1.47	0.024
C14:0	1.41 ± 0.57	1.39 ± 1.0	0.494
C15:0	0.78 ± 0.89	0.56 ± 0.84	0.362
C16:0	28.2 ± 2.97	29.5 ± 3.46	0.207
C18:0	0.46 ± 2.36	0.16 ± 0.66	0.855
C 20:0	0.02 ± 0.08	nd	0.375
C22:0	0.04 ± 0.22	nd	0.374
C16:1 n-7	3.94 ± 1.11	3.79 ± 1.16	0.731
C17:1 n-7	0.05 ± 0.22	0.09 ± 0.23	0.548
C18:1 n-7	1.41 ^A^ ± 2.09	0.05 ^B^ ± 0.25	0.005
C18:1 *cis* n-9	38.8 ± 5.62	39.6 ± 5.10	0.579
C18:1 *trans* n-9	11.1 ± 3.13	10.3 ± 4.56	0.333
C18:3 n-3	0.03 ± 0.10	0.04 ± 0.16	0.955
C22:6 n-3	nd	0.03 ± 0.11	0.270
C18:2 n-6	10.9 ^b^ ± 3.0	12.5 ^a^ ± 2.15	0.022
C18:3 n-6	0.02 ± 0.08	nd	0.375
C20:4 n-6	2.13 ^A^ ± 1.53	0.18 ^B^ ± 0.56	0.001
ΣSFA	31.5 ± 3.58	32.8 ± 4.50	0.297
ΣMUFA	56.3 ± 5.04	53.9 ± 4.79	0.338
ΣPUFA n-3	0.03 ± 0.10	0.06 ± 0.28	0.781
ΣPUFA n-6	13.0 ± 4.06	12.7 ± 2.15	0.977
ΣPUFA	13.0 ± 4.16	12.8 ± 2.07	0.939
ΣUFA	68.4 ± 3.68	66.6 ± 4.38	0.144

Mean values in rows marked with different letters differ significantly: ^a,b^ (*p* ≤ 0.05); ^A,B^ (*p* ≤ 0.01); nd—not detected.

**Table 3 animals-13-03298-t003:** Lipid quality indexes in *longissimus lumborum* ($\bar{x}\;\pm \;\mathrm{SD}$).

Indexes *	Control	Bokashi	*p*-Value
	(n = 24)	(n = 26)	
ΣMUFA/ΣSFA	1.79 ± 0.31	1.69 ± 0.42	0.307
ΣPUFA/ΣSFA	0.36 ± 0.10	0.40 ± 0.06	0.191
ΣUFA/ΣSFA	2.21 ± 0.34	2.09 ± 0.46	0.251
PUFA Σ n-3/Σ n-6	0.003 ± 0.01	0.006 ± 0.02	0.539
AI	1.04 ^b^ ± 0.99	1.73 ^a^ ± 1.52	0.032
TI	0.89 ± 0.15	0.94 ± 0.17	0.268
SI	2.52 ± 0.89	2.48 ± 0.48	0.906
UI	137.4 ± 7.46	133.4 ± 8.86	0.105
h/H	1.77 ± 0.24	1.73 ± 0.40	0.584
PI	67.0 ± 3.85	66.8 ± 4.96	0.270
DI (16)	12.2 ± 3.28	11.4 ± 3.16	0.649
DI (18)	99.1 ± 4.52	99.7 ± 1.51	0.591
TDI	1.82 ± 0.29	1.77 ± 0.42	0.604
EI	54.8 ± 4.72	54.3 ± 5.11	0.727
NVI	2.25 ± 2.33	1.89 ± 0.73	0.809
HPI	0.92 ± 0.09	0.91 ± 0.08	0.809
NR	0.50 ± 0.07	0.52 ± 0.10	0.448

Mean values in rows marked with different letters differ significantly: ^a,b^ (*p* ≤ 0.05). * See Table 1.

**Table 4 animals-13-03298-t004:** Fatty acid profile (% of total fatty acids) of pork backfat.

Fatty Acids	Control	Bokashi	*p*-Value
(%)	(n = 24)	(n = 26)	
C12:0	1.62 ± 3.79	0.34 ± 0.53	0.202
C14:0	1.39 ± 0.62	1.54 ± 0.39	0.256
C15:0	0.19 ± 0.81	0.07 ± 0.05	0.775
C16:0	26.4 ± 3.65	25.2 ± 1.85	0.282
C17:0	0.44 ± 0.82	0.44 ± 0.20	0.360
C18:0	12.7 ± 3.81	12.8 ± 2.01	0.464
C 20:0	0.07 ± 0.11	0.05 ± 0.07	0.429
C16:1 n-7	2.73 ± 0.97	2.34 ± 0.80	0.376
C17:1 n-7 cis	0.33 ± 0.15	0.37 ± 0.15	0.421
C18:1 n-7	2.78 ± 1.26	2.95 ± 0.30	0.811
C18:1 cis n-9	42.2 ^A^ ± 3.91	39.7 ^B^ ± 2.14	0.004
C18:1 trans n-9	0.14 ± 0.14	0.20 ± 0.07	0.697
C18:3 n-3	0.60 ^B^ ± 0.35	1.32 ^A^ ± 0.20	0.001
C20:5 n-3	0.08 ± 0.14	0.05 ± 0.10	0.308
C22:6 n-3	0.07 ± 0.19	0.14 ± 0.22	0.192
C18:2 n-6 cis	7.10 ^B^ ± 2.33	10.6 ^A^ ± 2.19	0.001
C18:3 n-6	0.05 ± 0.07	0.06 ± 0.09	0.444
C20:3 n-6	0.48 ^B^ ± 0.34	0.98 ^A^ ± 0.39	0.001
C20:4 n-6	0.16 ^B^ ± 0.15	0.37 ^A^ ± 0.19	0.001
ΣSFA	42.8 ± 6.18	40.4 ± 3.55	0.177
ΣMUFA	48.1 ^A^ ± 4.01	45.2 ^B^ ± 2.76	0.004
ΣPUFA n-3	0.76 ^B^ ± 0.50	1.51 ^A^ ± 0.66	0.001
ΣPUFA n-6	7.78 ^B^ ± 2.58	12.0 ^A^ ± 3.15	0.001
ΣPUFA	8.54 ^B^ ± 2.94	13.4 ^A^ ± 2.54	0.001
ΣUFA	56.7 ± 5.80	58.7 ± 3.17	0.237

Mean values in rows marked with different letters differ significantly: ^A,B^ (*p* ≤ 0.01).

**Table 5 animals-13-03298-t005:** Lipid quality indexes of pork backfat ($\bar{x}\;\pm \;\mathrm{SD}$).

Indexes *	Control	Bokashi	*p*-Value
	(n = 24)	(n = 26)	
Σ MUFA/ΣSFA	1.14 ± 0.42	1.13 ± 0.15	0.771
Σ PUFA/ΣSFA	0.28 ^B^ ± 0.14	0.44 ^A^ ± 0.11	0.001
Σ UFA/Σ SFA	1.36 ± 0.53	1.47 ± 0.21	0.162
PUFA Σ n-6/Σ n-3	8.16 ± 4.28	6.96 ± 2.36	0.938
PUFA Σ n-3/Σ n-6	0.09 ^b^ ± 0.05	0.13 ^a^ ± 0.04	0.011
AI	1.83 ± 2.92	1.10 ± 0.560	0.408
TI	1.27 ± 0.39	1.20 ± 0.21	0.753
SI	4.49 ^A^ ± 1.81	2.97 ^B^ ± 0.74	0.009
UI	105.4 ± 24.6	111.0 ± 5.8	0.275
h/H	1.83 ± 0.68	2.0 ± 0.22	0.114
PI	53.8 ± 12.7	59.3 ± 3.89	0.369
DI (16)	9.53 ± 3.60	8.55 ± 2.95	0.705
DI (18)	73.9 ± 17.3	75.7 ± 3.32	0.340
TDI	1.13 ± 0.44	1.08 ± 0.14	0.925
EI	62.6 ± 14.0	65.6 ± 1.18	0.284
NVI	13.6 ± 4.54	14.3 ± 1.90	0.213
HPI	1.21 ± 0.28	1.27 ± 0.08	0.286
NR	0.56 ± 0.15	0.54 ± 0.05	0.757

Mean values in rows marked with different letters differ significantly: ^a,b^ (*p* ≤ 0.05); ^A,B^ (*p* ≤ 0.01); * See Table 1.

## Data Availability

The data presented in this study are available on request from the corresponding author.

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
