# Peer review of "Fatty Acid Profile and Lipid Quality Indexes of the Meat and Backfat from Porkers Supplemented with EM Bokashi Probiotic"

_animals, 2023, doi:10.3390/ani13203298_

Round 1
Reviewer 1 Report
This manuscript studied the effect of probiotic supplementation on the fatty acid profile in pig, and provided information on the application of probiotic for healthy pig meat production. Suggestions and comments as following:
Introduction
The introduction provides a lengthy introduction to the relationship between probiotics and health, but there is insufficient research progress on the impact of probiotics on animal production.
Is the fatty acid profile in Table 1 derived from the longissimus lumborum muscle or IMF?
Is the fatty acid content in Table 1 a percentage of longissimus lumborum muscle or a percentage of total fatty acids?
Is there any data on backfat thickness?
minor editing
Author Response
We thank the Reviewer for all comments that will make our revised manuscript more understandable to the reader. Below are the responses to the remarks contained in the review.

Reviewer 2 Report
Title: “Fatty acid profile and lipid quality indices of the meat and backfat from porkers supplemented EM Bokashi Probiotic”
The study aimed to assess the effect of supplementation of pig diet with the probiotic 106 EM®Bokashi on the fatty acid profile of intramuscular fat (IMF) of their muscles and back-107 fat, as well as on lipid quality indicators informing about the possible impact on consumer 108 health.
The results of this study are interesting because its approach is associated with nutrition recommendations and human health beyond an animal-productive approach. However, the presentation of the results becomes diffuse when the results and discussion sections are put together. Besides, the form of presenting results in tables is unremarkable. Alternative analyses and figures, such as clusters, could be explored. A table showing the grouping of fatty acids and lipid quality indices in the section on materials and methods would be appropriate.
Furthermore, in the discussion of this work, some concepts should be defined and included in the introduction before the results section to understand better the analysis done. Finally, the conclusion needs to be clarified and generates a certain contradiction. First, it is indicated that “The changes in the fatty acid profile and lipid indices in backfat are more favorable to consumer health than in meat.” Then it is indicated that “EM®Bokashi probiotic at a dose of 3 g/kg of feed in the last stage of pig production does not have an overall positive effect on the fatty acid profile of the meat”. “Since consumers more often consume meat than pork fat.”
Specific comments:
Line 115: The dosage of EM®Bokashi probiotic is not clear and apparently without scientific support
Line 119: Please specify environmental conditions
Line 159: define the acronym AOAC
Line 170: Define the acronym FAMEs
Line 192: Please present the information in a table in a more organized manner than it is currently shown.
Line 208: Concepts are mentioned without a prior definition (e.g., Atherogenicity index)
Line 240: The results and discussion section becomes very long; they could be placed separately and independently
Lines 258-259: Add cite of this paragraph: “Simultaneously, its presence is welcomed because it is associated with better taste, tenderness, aroma, and juiciness.”
Lines 447, 486, and 505: In all tables of results, it is suggested to add “*” to significant p-values
Author Response

(The authors gave the same response as above.)

Reviewer 3 Report
Fatty acid profile and lipid quality indices of the meat and backfat porkers supplemented EM®Bokashi Probiotic
Dear Authors,
the manuscript is well prepared and describe effect of the used commercial probiotic on the fatty acid profile and lipid quality indices of the meat. There are two main things which need to be consider: application of trade name of probiotic in title of manuscript/article and description of details in statistical analysis (normality test, and homogeneity of variance).
Below I add some suggestions helpful during this process:
Line 3
In title of manuscript is: “…EM®Bokashi Probiotic”, in this case registered trademark sign should be use after Bokashi name: EM Bokashi® Probiotic.
But maybe better will be to left only EM Bokashi Probiotic like in Rybarczyk et al. (2021, article in Animals Journal), because in case fatty acid profile of the meat and lipid quality indices there are not many significant differences, what can suggest that the probiotic have main effect in case performance of porkers, but not have bigger influence for lipid quality and fatty acid profile. Additionally in case the statistical analysis the logarithmic transformation was needed, what suggest quite big dispersion of data before transformation, and after transformation not much significant differences was found. Perhaps the probiotic effect ewill be better visible in case SCFA synthetised by microorganisms in the colon and level of immunoglobulin in blood serum as an effect of GALT action in porkers.
Line 17
EM Bokashi®, if kept the trade name.
Line 112
2. Materials and Methods subsection must be transferred to line 111.
Line 233
Better will be to delete words: or approach, because in this case p-value in Shapiro-Wilk test could not be in the critical area and be lower than 0.05 (it must exceed 0.05, and be in confidence interval what allow to accept H0 described as a: lack of difference between analysed distribution of data and normal distribution).
Of course information about Shapiro-Wilk normality test for each treatment is needed, because if data in treatment will not have the normal distribution, the non-parametrical U-Mann-Whitney test must be used, also information about homogeneity of variances in treatments is needed.
Very conservative tests was chosen (one-way ANOVA, and Tukey’s post hoc test), maybe it will be worth to consider using simpler t-test for two treatments (in case lack of homogeneity of variances – Welch’s t- test).
Line 280, 291 and 356
The same like in line 17
Line 375, 486 and 505
Maybe it is possible to centre values for mean and standard deviation in column 2 and 3.
Line 423, 490
The same like in line 17
Line 506
In text of manuscript is: A,B (p ≤ 0.05), must be: A,B (p ≤ 0.01)
Line 509, 515
The same like in line 17
Lines 536- 749
Dots in abbreviation of Journal name needed
Journal name in abbreviated form needed in lines: 558, 612, 637, 654, 657, 666, 697, 682, 697, 707, 724
Line 636
Sari et al. (2015). There is the English title :”Effects of different fattening systems on technological properties and fatty acid composition of goose meat”, and article is in an English language, that is why it will be better taking into a consider this title, despite of second in the German language.
Author Response

(The authors gave the same response as above.)

Round 2
Reviewer 2 Report
The authors performed some of the modifications suggested
Author Response
We have made changes to lines 43-45; 75-77; 84; 119; 120; 121; 129-131; 155-157; 163; 174-175; 178-183; 186-190; 202-206; 245-247; 255-264; 375-
377; and in Table 1 in two first lines.